# Matched Filter Interpretation of CNN Classifiers with Application to HAR

**DOI:** 10.3390/s22208060

**Published:** 2022-10-21

**Authors:** Mohammed M. Farag

**Affiliations:** 1Electrical Engineering Department, College of Engineering, King Faisal University, Al-Ahsa 31982, Saudi Arabia; mfarag@kfu.edu.sa; 2Electrical Engineering Department, Faculty of Engineering, Alexandria University, Alexandria 5424041, Egypt; mmorsy@alexu.edu.eg

**Keywords:** machine learning, convolutional neural network, interpretable neural network, matched filter, human activity recognition

## Abstract

Time series classification is an active research topic due to its wide range of applications and the proliferation of sensory data. Convolutional neural networks (CNNs) are ubiquitous in modern machine learning (ML) models. In this work, we present a matched filter (MF) interpretation of CNN classifiers accompanied by an experimental proof of concept using a carefully developed synthetic dataset. We exploit this interpretation to develop an MF CNN model for time series classification comprising a stack of a Conv1D layer followed by a GlobalMaxPooling layer acting as a typical MF for automated feature extraction and a fully connected layer with softmax activation for computing class probabilities. The presented interpretation enables developing superlight highly accurate classifier models that meet the tight requirements of edge inference. Edge inference is emerging research that addresses the latency, availability, privacy, and connectivity concerns of the commonly deployed cloud inference. The MF-based CNN model has been applied to the sensor-based human activity recognition (HAR) problem due to its significant importance in a broad range of applications. The UCI-HAR, WISDM-AR, and MotionSense datasets are used for model training and testing. The proposed classifier is tested and benchmarked on an android smartphone with average accuracy and F1 scores of 98% and 97%, respectively, which outperforms state-of-the-art HAR methods in terms of classification accuracy and run-time performance. The proposed model size is less than 150 KB, and the average inference time is less than 1 ms. The presented interpretation helps develop a better understanding of CNN operation and decision mechanisms. The proposed model is distinguished from related work by jointly featuring interpretability, high accuracy, and low computational cost, enabling its ready deployment on a wide set of mobile devices for a broad range of applications.

## 1. Introduction

A convolutional neural network (CNN) is a prominent machine learning (ML) architecture inspired by the natural visual perception mechanism of the human visual cortex. The foundation of convolutional neurons has been established since the 1950s [1]. In 1989 LeCun et al. [2] established the modern framework of CNN. They developed a multilayer CNN, namely LeNet-5, for handwritten digit classification that can be trained with the backpropagation algorithm. Since then, many works investigated CNN architectures, applications, and training methods, however, due to the lack of large datasets and sufficient computing power at that time, these works did not achieve the expected results [1].

It was not until 2012 that AlexNet was advanced for computer vision tasks for the first time [3]. AlexNet is a multilayer CNN that uses GPUs for model training and comprises a variety of kernels. The performance of AlexNet, the winner of ImageNet Challenge 2012, superseded all existing non-neural model rivals by a considerable margin. With the success of AlexNet, many milestones have been advanced to improve CNN performance, such as VGGNet [4], ZFNet [5], GoogleNet [6], and ResNet [7]. A typical trend of the CNN architecture evolution is developing deeper networks seeking performance improvements. By increasing the network depth, model expressiveness increases, leading to achieving better results. However, the model complexity also increases, which makes CNN more difficult to optimize and prone to overfitting. During the last decade, various methods have been proposed to overcome these challenges [1].

In addition to their outstanding performance, another prominent feature of CNNs is their automatic feature extraction capabilities. While conventional ML methods usually perform feature engineering and preprocessing procedures to extract handcrafted features, which are not only suboptimal but also computationally intensive, CNN models can automatically extract representative features directly from the raw data of the problem at hand to maximize classification accuracy. Automatic feature extraction is the key enabler for significantly improving model performance without the need to field expertise, which makes CNNs very attractive for solving complex problems [1].

Time series are a sequence of 1D data points which are collected by recording a set of observations of a specific activity chronologically. The ubiquity and impact of time series data accompanied by the prevalence and continual advancements of modern sensors open unlimited research frontiers for data scientists. However, time series are characterized by large data sizes, high dimensionality, and nonstationarity, which makes time series analysis using ML algorithms a challenging task. Due to their significant leaps in improving ML models for computer vision and 2D grid data such as images, CNNs have also been investigated for time series. CNNs have been used in various supervised and unsupervised time series problems including classification, regression, and forecasting. In a relatively short time, 1D CNNs have become popular, with state-of-the-art performance in various time series ML applications such as electrocardiogram (ECG) classification, human activity recognition (HAR), and industrial fault diagnostics and predictive maintenance [8].

On the other hand, despite their significant achievements in ML, deep learning (DL) and CNN models are commonly used as black box models because their internal operation and decision mechanisms are not explicitly understood by humans. The lack of interpretability has become the main barrier to using DL in mission-critical applications [9]. For mission-critical tasks, it is essential to verify that the model heightened accuracy results from the use of a proper problem representation, not from the exploitation of artifacts in data. Furthermore, model robustness and interpretability are closely related because interpretability enables identifying potential vulnerabilities of complicated ML models [9].

In this work, we present a matched filter (MF) interpretation of CNN classifiers for time series. An MF is an optimal linear filter for signal detection in the presence of additive white Gaussian noise (AWGN) by maximizing the signal-to-noise ratio at the receiver side [10]. MFs are widely used in signal processing and wireless digital communications for optimal signal detection. The block diagram of the MF receiver used in digital communication systems is illustrated in Figure 1. An MF works by correlating the received noisy signal with the known signal template, sampling the maximum output of the correlator, and making the decision if the received noisy signal matches the template or not.

We develop an MF-based CNN model for time series classification using a stack of convolutional, pooling, and fully connected (FC) layers and provide an analytical intrinsic interpretation of the developed model. Raw time series data are fed to the model with no need to feature engineering. A 1D convolutional (Conv1D) layer is used as an MF correlator, a GlobalMaxPooling (GMP) layer is used as a maximum sampling block, and an FC layer with softmax activation is used as the decision block for computing the class probabilities.

Unlike the trend of increasing the model depth to increase its expressiveness [1], the proposed model is shallow in depth with a wide receptive field (wide kernels), which preserves the model expressiveness and performance merits without increasing the model complexity. The presented interpretation is experimentally validated using a carefully developed synthetic time series dataset. The proposed interpretation helps develop a better understanding of the CNN learning process. Moreover, the conducted experiments draw some useful guidelines for developing efficient CNNs. Furthermore, the presented interpretation enables developing superlight highly accurate classifier models that meet the tight requirements of edge inference. Edge inference is an emerging research direction that promotes using embedded devices to perform model computations locally instead of relying on cloud computing. Edge inference addresses the latency, availability, privacy, and connectivity concerns of the commonly deployed cloud inference.

Afterward, the model is applied to the HAR problem for further validation of the model performance on real datasets. Three commonly used datasets, namely UCI-HAR, MotionSense, and WISDM-AR are used for model training and testing to validate the model performance and generalization capabilities. The developed HAR models are tested and benchmarked on a cloud machine and an android edge device. The achieved results are superior compared with state-of-the-art results, especially in terms of the classifier’s real-time performance. Moreover, an android HAR application is developed using the MF CNN classifier and tested on an android smartphone. The proposed model features three distinguishing characteristics: high accuracy, low computational cost, and interpretability, enabling model deployment on a broad set of edge devices for a wide variety of applications.

The contributions of this work are:Providing a clear interpretation of CNN classifiers as MF and presenting an experimental proof of concept to support this interpretation.Presenting a superlight highly accurate CNN classifier model for time series applications that suits both cloud and edge inference approaches.Applying the developed model to the renowned HAR problem and achieving outstanding results compared with the state-of-the-art modelsTesting and benchmarking the MF CNN classifier on an edge device and developing an android HAR application.

The remainder of this paper is organized as follows. In Section 2, a brief background on ML interpretation methods and HAR related work is presented. The MF interpretation of CNN classifiers and its experimental validation are advanced in Section 3. Application of the MF CNN classifier to HAR, along with the used tools and methods, is introduced in Section 4. Performance results of the HAR MF CNN classifier and a comparison between the proposed model and related HAR methods are advanced in Section 5. Conclusions and future work are portrayed in Section 6.

## 2. Literature Review

### 2.1. CNN Interpretation Background

Methods for interpreting ML models are classified according to different criteria. First, the intrinsic or post hoc classification is based on whether the interpretation is conducted by restricting the complexity of the interpreted ML model (intrinsic or ad hoc) or by using post-training methods to analyze the trained model (post hoc). Intrinsic interpretation is feasible for ML models that can be considered interpretable by structure such as short decision trees or sparse linear models. Post hoc interpretation refers to applying interpretation methods, such as visualizing feature maps and model weights after model training to understand the model operation and decision mechanism. The second classification norm is the model-specific or model-agnostic criterion. Model-specific interpretation tools are limited to specific model classes, such as tools that only work for interpreting a specific ML model. Model-agnostic tools can be applied to any ML model after the model has been trained (post hoc). These methods commonly work by analyzing feature input and output pairs without accessing model internals, such as weights or structural information. Third, the local or global classification criterion is based on the interpretation method ability to explain an individual prediction or the entire model behavior [11].

There are several works that present interpretations of CNNs, however, most of them target 2D CNNs [9,12]. Fortunately, there are some works that investigate the interpretation of 1D CNNs. Srinivasamurthy [13] carefully crafted time series datasets and observed the CNN model weights in both time and frequency domains. He proposed a frequency domain interpretation of CNN filters, studied the effect of dropout regularization, and provided guidelines for choosing the kernel length based on the input data.

Pan et al. [14] proposed an interpretable 1D CNN for a biomedical time series classification application. For interpreting the classification results, the FC output layer is replaced with a global average pooling (GAP) layer that generates one feature map for each corresponding class in the last convolution layer. In this work, GAP is employed in the FC output layer of the 1D CNN model for generating a class activation map (CAM). The GAP layer can highlight distinguishable parts of the input waveform that contributed to the classification result produced by the CNN classifier. The GAP layer computes the spatial average of the feature map of each kernel at the last convolutional layer, such that the importance of the input waveform sections can be differentiated in classification by applying the weights of the FC output layer to the convolutional feature maps. Unfortunately, using the GAP layer reduces the number of model parameters greatly, which consequently compromises the classifier performance.

Similarly, Wang et al. [15] proposed using CNNs with GAP and CAMs to distinguish class-specific regions in the input waveform. CAMs can disclose contributing subsequences in raw input data that result in a specific label, which enables visualizing the predicted class scores on time series data and highlighting the discriminative subsequences that led to such a classification result. Unfortunately, using GAP instead of the FC layer degrades the classifier performance, which requires applying some changes to the CNN model to compensate for the FC layer removal effect. Such alterations in the model would also result in rendering the interpretation results less useful as the interpreted model has been altered.

Stankovic and Mandic [16] employed the MF paradigm as a mathematical means to demystify the operation of CNNs. A close inspection of the convolutional layer within CNNs reveals a direct link with MFs in finding features (patterns) in data. Such a framework facilitates linking matched filtering to feature identification and establishes a clear approach for understanding the information flow in CNNs’ learning and their optimal parameter selection mechanism. This interpretation is accompanied by an evaluated example and supported by detailed numerical outputs and visualizations. This perspective serves as a basis for a tutorial on the operation of CNNs. The interpretation provided by this work matches our interpretation of the CNN classifier. Unfortunately, the CNN MF interpretation presented in this work was not validated on a real dataset to show the CNN model performance and limitations in practical applications.

In our previous work [17], we presented a finite impulse filter (FIR) interpretation of the Conv1D layer and exploited this interpretation to develop a self-contained short-time Fourier transform (STFT)-based CNN ECG classifier. The Conv1D layer filters are designed to implement a filter bank that is used to extract the time–frequency spectrogram of the input ECG signal inside the model. The Conv1D output feature maps are reshaped into a 2D heatmap image and fed to a 2D CNN model for classification. In this work, the Conv1D layer filter activation and heatmaps are visualized to show the model ability to extract time–frequency images of the input signals that match STFT spectrograms.

In [18], we presented an MF interpretation of the Conv1D layer and exploited this interpretation to develop a light CNN model for interpatient ECG classification and arrhythmia detection at the edge using a single-lead (a univariate time series). An average-based MF template is derived for each ECG class in the training set and preassigned to the Conv1D layer kernels. A stack of a Conv1D layer followed by a GMP layer acts as a typical MF for identifying the heartbeat class. A very important conclusion drawn from this work is that interpretable usage of neural networks leads to develop better models.

In this work, we aim to support our MF interpretation of CNNs with an experimental proof of concept and use this interpretation to better understand the CNN operation and optimal parameter selection. The presented interpretation method can be classified as both ad hoc (intrinsic) and post hoc, global, and model-specific methods. The proposed interpretation is specific to the CNN model presented in this work (yet it can be later generalized) and can be globally used to explain the entire model behavior. The model structure follows MFs (ad hoc), and the model feature maps will be visualized (post hoc) for interpreting the model operation. The proposed interpretation approach does not require any alteration to the model structure such as in GAP-based methods.

### 2.2. Human Activity Recognition Related Work

HAR has become a very active research area due to its wide range of applications in elderly care, healthcare, smart homes, athletics, and abnormal activity monitoring. According to the World Health Organization (WHO) fact sheet reported in 2021, more than 1 billion individuals have some sort of disability [19]. Currently, there are insufficient facilities to address the needs of persons with disabilities. One of them is the need for a companion to watch their activities. The activities of people with disabilities can be automatically watched using HAR to safeguard them from injury, danger, or accidents.

HAR aims to accurately classify human physical activities using raw time series signals or video records acquired through specialized sensors or video cameras. The ubiquity of wearable devices and mobile phones which are equipped with a plethora of sensors has created new research opportunities and applications for HAR. Due to its significant importance in various applications, and its close relevance to time series classification, we investigate applying the proposed MF CNN classifier to the sensor-based HAR problem.

In this section, modern, related work of HAR using ML is presented. Most modern ML methods for HAR are based on CNNs, long short-term memory (LSTM) recurrent neural networks (RNNs), gated recurrent unit (GRU) RNN, or a hybrid of these methods. The most widely used datasets for model training are the Wireless sensor data mining (WISDM) [20] and University of California Irvine (UCI) [21] HAR datasets.

Ignatov [22] proposed a user-independent DL-based approach for online HAR. A shallow 1D CNN model is used for local feature extraction using raw sensor data. Global statistical features about the time series are also fed to the model. The impact of time series length on model performance is investigated in this work. The proposed approach is evaluated on the WISDM-AR and UCI-HAR datasets The results show that the proposed model achieves acceptable performance using low computational cost. However, extracting statistical features from sensor data requires using feature engineering and preprocessing stages and, consequently, additional computational cost.

Xia et al. [23] proposed a deep neural network (DNN) that combines convolutional layers with LSTM. In the proposed model, raw sensor data are fed to a two-layer LSTM followed by convolutional layers. A GAP layer is used to replace the FC layer after convolution to reduce the number of parameters of the model, and a batch normalization (BN) layer is added after the GAP layer to speed up model training. The model performance is evaluated on the UCI, WISDM, and OPPORTUNITY datasets. The proposed model exhibits acceptable performance using a small number of parameters. However, the model depth and layer diversity increase the computational cost of the proposed classifier.

Nafea et al. [24] proposed a CNN model with variable kernel dimensions along with a bidirectional LSTM (BiLSTM) layer to capture time series features at various resolutions. The model is composed of hybrid Conv1D and BiLSTM stacks, each comprising multiple cascaded layers, and is fed with raw sensor data. The UCI-HAR and WISDM-AR are used for model training and testing. Results indicate that the proposed scheme is efficient in improving HAR. However, using a hybrid DNN model with diverse layers increases the model complexity of the proposed network. Moreover, LSTM RNNs suffer from increased computation time, limiting their applicability to edge inference.

Yin et al. [25] proposed a 1D CNN-BiLSTM parallel model with an attention mechanism. Raw sensor data are segmented and fed into a 1D CNN-BiLSTM parallel layer to accelerate feature extraction. Extracted feature weights are redistributed by the attention mechanism and integrated into complete features. The classification results are computed using an FC layer. The performance is evaluated on public UCI-HAR and WISDM-AR datasets, and the achieved results are promising. However, using a hybrid DNN model with diverse layers increases the model complexity and computational cost. Furthermore, LSTM RNNs suffer from increased computation time, limiting their applicability to edge inference.

Tan et al. [26] proposed an ensemble learning algorithm (ELA) to perform HAR using smartphone sensors. The proposed ELA model combines a GRU layer, a CNN stacked on the GRU, and a multilayer FC DNN. The DNN input samples compose an extra feature vector consisting of both time-domain and frequency-domain parameters. The FC DNN was used to fuse three models and performs the classification task. The proposed approach is evaluated on the UCI-HAR dataset, and the achieved results are comparable to related works. However, using a DNN model with diverse layers increases the model complexity and computational cost of the proposed network. Additionally, GRU RNNs suffer from increased computation time, limiting their applicability to edge inference.

Pushpalatha and Math [27] proposed a hybrid deep CNN model for HAR. A stack of 1D CNNs is used for feature extraction, after which a GRU stack is used to capture the long-term dependency between the different actions. The model is evaluated on the UCI-HAR dataset, and the achieved results are comparable to the state-of-the-art HAR results. Unfortunately, the model is only tested on a single dataset, which does not establish the model generalization capabilities. Moreover, such a deep model with hybrid layers is not suitable for edge deployment due to the increased computational cost.

Sikder et al. [28] proposed a HAR model using a two-channel CNN which is fed with frequency and power features of the sensor data. The model is tested on the UCI HAR dataset, and the achieved results are acceptable. Unfortunately, the model is only tested on a single dataset, which does not establish the model generalization capabilities. Furthermore, using such a deep model increases the computational cost of the model.

Luwe et al. [29] proposed a hybrid DNN that comprises a 1D CNN with a 1D BiLSTM model and is fed with raw sensor data. The 1D CNN extracts high-level representative features from raw sensor data, which are fed to the BiLSTM layer that encodes the long-range dependencies between features by gating mechanisms. The model is evaluated on the UCI-HAR and MotionSense datasets. The performance evaluation reveals that the proposed model outperforms the compared existing methods. However, using a DNN model with hybrid layers increases the model complexity and computational cost of the proposed classifier. More specifically, RNNs such as LSTM suffer from increased computation time, limiting their applicability to edge inference.

Ronald et al. [30] proposed iSPLInception, a DL model inspired by the Inception-ResNet architecture from Google [31]. Inception modules are used for building very deep and wide CNN models. In each Inception module, convolutions are conducted in parallel with different kernel sizes, and the output from these parallel operations is concatenated. The proposed model is evaluated on four public HAR datasets from the UCI ML repository. The experiments and result analysis indicate that the proposed iSPLInception model achieves outstanding performance in HAR. However, such a deep model is not the best fit for edge inference, which requires smaller models with a reduced computational cost.

Sannara EK [32] presented HAR Transformer (HART), a lightweight, sensorwise transformer architecture that has been specifically adapted to the domain of sensory data. The model is based on the successful vision transformer model, and it has been extended to MobileHART. This architecture is also compared against classical lightweight CNN and CNN-LSTM and outperformed them on several datasets. The performance of various HART architectures has been evaluated in heterogeneous environments and showed that proposed models can better generalize to different sensing devices or on-body positions. However, the model number of parameters is greater than 1 million, which raises serious concerns about its suitability for edge inference on tightly resource-constrained devices.

On the other hand, there are some works that address self-supervised, semisupervised, and transfer learning of HAR. Tang et al. [33] presented SelfHAR, a semisupervised model that leverages unlabeled HAR datasets to complement small labeled datasets. SelfHAR combines teacher–student self-training to extract the knowledge of unlabeled and labeled datasets to learn robust time series representations by predicting distorted versions of the input. SelfHAR is evaluated on various HAR datasets and showed acceptable performance for both supervised and semisupervised approaches. Unfortunately, results achieved by this model fall behind their supervised learning counterparts by a considerable margin.

Rahimi Taghanaki et al. [34] proposed self-supervised learning for HAR with smartphone accelerometer data. The proposed solution consists of two steps. The representations of unlabeled input signals are learned by training a deep CNN to predict a segment of accelerometer values. This model leverages past and present motion in the x and y dimensions, as well as past values of the z-axis to predict values in the z dimension. Next, the convolution blocks are frozen, and the weights to the downstream network are transferred. For computing classification results, a number of FC layers are added to the end of the frozen network, and the added layers are trained with labeled accelerometer signals to learn to classify human activities. The performance of the proposed method has been evaluated on three datasets: UCI HAR, MotionSense, and HAPT. The achieved performance of this model is acceptable compared with related models. Unfortunately, results achieved by this model fall behind their supervised learning counterparts by a significant margin.

Taghanaki et al. [35] proposed a self-supervised learning method for HAR using smartphone accelerometer data that reduce reliance on class labels. Cross-dataset transfer learning is performed such that the model pretrained on a particular dataset can be applied to other datasets after a small amount of fine-tuning. Two separate pipelines are developed: a time–frequency domain pipeline using STFT scalograms, and a time-domain pipeline. The two streams are then fused to provide the final classification results. Self-supervised contrastive learning is used to train each of these streams. The performance of the proposed solution is evaluated on three publicly available datasets. The achieved results demonstrate that this solution outperforms the compared related works. Unfortunately, results achieved by this model fall behind their supervised learning counterparts by a considerable margin.

Table 1 summarizes methods and limitations of HAR-related work presented in this section. The main purpose of this work is to advance a clear interpretation of CNN classifiers and develop a highly accurate computationally inexpensive model for time series classification on edge devices. The main challenge is meeting the classification accuracy requirements using a resource-constrained edge device. Most related time series classification works follow the trend of increasing the model depth and using hybrid layers to increase the model accuracy at the expense of increasing the model complexity and computational load, rendering them less suitable for edge inference. Furthermore, most related works neither present real-time benchmarking results nor performance analysis of the developed models. Many important results, such as the model size, number of parameters, memory usage, and computation time, are neither presented nor optimized in the literature either. On the other hand, most related works do not present a clear interpretation of the proposed models, which limits their usage in mission-critical applications such as healthcare. In this work, we aim to address the above challenges and provide an interpretable, superlight, and highly accurate ML classifier for time series applications ready for edge deployment.

## 3. Matched Filter Interpretation of Convolutional Neural Network

A 1D CNN is a DL model for processing time series data that is inspired by the architecture of the human visual cortex and designed to learn spatial hierarchies of features automatically and adaptively, from low- to high-level sequences. A CNN classifier is a mathematical construct composed of three types of layers: convolutional, pooling, and FC layers. The first two layers, convolution and pooling, extract features, while the third, an FC layer, uses the extracted features for classification. A deep CNN classifier is built up using multiple cascaded stacks of convolutional and pooling layers to increase the model expressiveness power and feature extraction capabilities. Each 1D convolutional (Conv1D) layer contains a number of 1D filters, also called kernels, to extract several feature maps from the input signal. A Conv1D layer slides several kernels across a time series sequence to produce a 1D feature map per kernel. The shift amount is determined by the number of strides parameter. A bias parameter can be also used to fine-tune the kernel for better results. Activation functions are inserted to add nonlinearity to the CNN model. Each kernel will learn to detect a single sequential pattern of the kernel length.

Despite its name, the Conv1D layer does correlate the input to the layer with the layer kernel weights. The Conv1D layer output is expressed as follows [8]:(1)$$y_{k}^{l}=f\left(b_{k}^{l}+\sum_{i=1}^{N_{l-1}}w_{ik}^{l-1}x_{i}^{l-1}\right)$$
where $y_{l}^{k}$ is the layer output, f() is the activation function, $b_{l}^{k}$ is the bias of the *k*th neuron at layer *l*, $x_{i}^{l-1}$ is the output of the *i*th neuron at layer l−1, $w_{ik}^{l-1}$ is the kernel weight from the *i*th neuron at layer l−1 to the *k*th neuron at layer *l*, and $N_{l-1}$ is the size of the Conv1D kernel at layer l−1.

While the use of CNNs for feature extraction has become a de facto standard, interpretation of CNN operation is still an open question [16]. To this end, the MF theory is used to draw an interpretation of CNN operation. In signal processing, an MF is obtained by correlating a known signal, or template, with an unknown noisy signal to detect the presence of the template in the unknown signal. MFs are widely used in digital communications for optimal signal detection. The output of an MF is calculated by correlating the template of the signal to be detected with the unknown noisy signal and comparing the maximum correlation output to a precalculated threshold to make the signal detection decision as shown in Figure 1. The output of the MF correlator is defined as follows:(2)$$y\left[n\right]=x\left[n\right]☆h\left[n\right]=\sum_{i=1}^{N}x\left[i\right]h\left[n-i\right]=\sum_{i=1}^{N}x\left[n-i\right]h\left[i\right]$$
where y[n] is the MF output, x[n] is the input signal, h[n] is the template signal of the MF (also called filter taps), *i* is the time shift, and ☆ is the 1D correlation operator.

Comparing Equations (Equation 1) and (Equation 2) shows that the Conv1D operation is equivalent to the MF correlation operation, where the Conv1D filter kernel $k\left[n\right]=w_{k}^{l-1}$ is equivalent to the MF template signal h[n]. The shifting operation is performed by sliding the kernel and correlating it with the input signal samples for all values of *n* (for the number of Conv1D strides is set to 1). To complete the MF operation, a GMP layer can be used to select the maximum output of the Conv1D layer and perform the operation of the sampling device of Figure 1. Finally, an FC layer is used to perform the thresholding and decision operation of Figure 1 and maps the GMP layer outputs to the corresponding class probability outputs. The Conv1D layer comprises multiple filters with different learnable kernel weights. This stack of Conv1D, GMP layers, and FC layers works together as a typical MF with a template $h_{i}\left[n\right]=k_{i}\left[n\right]$, where *i* represent the *i*th kernel of the Conv1D layer.

Figure 2 shows the proposed model inspired by the MF interpretation of CNN classifiers. The model is composed of a stack of a Conv1D layer followed by BatchNormalization (BN), Tanh activation, GMP, and FC layers. BN is a regularization technique that normalizes a layer input by subtracting the minibatch mean and dividing it by the minibatch standard deviation, reducing the internal covariate shift and instability in distributions of layer activations. BN does not affect the MF operation, since all batch examples are uniformly normalized using the same values. Tanh activation is inserted in this model to add nonlinearity to the proposed model and increase the model expressiveness. The GMP layer outputs the maximum correlation of each feature map. It should be indicated that for a typical MF operation, the used pooling layer should be GMP, not GAP, because the learning process is directly influenced by the model topology, such that insignificant changes in the model topology would result in significant changes in the learned parameters and model operation. Outputs of the GMP layer are then fed to an FC layer with softmax activation to compute class probabilities. The model loss function is set to the categorical cross-entropy loss, which is typically used for multiclass classification problems.

The operation of the MF CNN classifier is explained as follows: Raw time series signals are fed to the model. The Conv1D MF kernels correlate the input signal with the layer kernels. The kernel weights will be automatically tuned up using the backpropagation algorithm to match the template signal representing a specific class and maximize the correlation output. Each kernel will learn a specific class template that is matched to the filter path to the class probability output via the GMP and FC layers. For a matched input signal, a single kernel will compute the autocorrelation between the signal and the corresponding MF kernel, while the remaining kernels will compute the cross-correlation between the signal and the unmatched kernel templates. If the autocorrelation maximum is much greater than the cross-correlation maximum (highly uncorrelated signals), the signal can be easily identified, whereas highly correlated signals can result in high cross-correlation values and, consequently, wrong predictions. The shift-invariant GMP layer will select the maximum output of all Conv1D filters, which are then fed to the FC output layer. The weights of the FC layer will be automatically tuned up during model training to minimize divergence between the predicted and ground-truth labels. The output layer weights will be tuned up such that signals belonging to a specific class are mapped to the corresponding class probability output to minimize the loss function.

The MF interpretation of CNN classifiers can be extended to deeper CNNs composed of cascaded Conv1D and MaxPooling layers. In a typical deep CNN, multiple convolutional layers, each comprising multiple filters followed by pooling layers, are hierarchically stacked to extract features and patterns of the training set with the aid of the backpropagation algorithm. A stack of FC layers is then used to produce the model output according to the model function of either classification or regression. For classification problems, convolutional kernels act as MFs with learnable weights that are automatically tuned up during model training to minimize the model loss function by maximizing correlation with specific discriminative features learned from the training set examples. Therefore, visualizing convolutional layer feature maps in 2D CNNs for computer vision demonstrates similarities with the dataset features; for example, visualizing feature maps of CNNs used for face recognition shows parts of the face such as the eyes and nose.

### 3.1. Experimental Proof of Concept

The presented interpretation of the CNN classifier is intrinsic or ad hoc, since the topology of the proposed model is based on the MF architecture. However, in this section, we provide a post hoc experimental proof of concept of the MF interpretation of CNN classifiers by training and testing the proposed model on a carefully selected synthetic dataset under several conditions and visualizing the model learned weights to support our MF interpretation of CNN.

#### 3.1.1. Synthetic Dataset

The approach used to generate the synthetic time series dataset is using a number of template signals with specific patterns to randomly generate a set of examples following the same pattern under the effect of random amplitude and time shift variations. A set of *N* time-limited template signal segments with various waveforms are generated using different mathematical functions. The number of samples (duration) of each signal is arbitrarily set to 64. Figure 3a shows the generated signals for N=4.

Each signal represents a template for randomly generating hundreds or thousands of signals, according to the dataset size, by the means of randomly shifting the signal segment along the time access and adding random noise to the signal. The number of examples generated from each template signal $N_{T}\left[i\right]$, where *i* represents the template signal index, is set to be a controlled variable to change the distribution of dataset examples from balanced to unbalanced. The duration (number of samples) of the dataset examples is set to 128 samples, which is double the template signal duration. For each template signal, $N_{T}\left[i\right]$ examples are generated by shifting the 64-sample template segment randomly using a random integer varying between 0 and 64 and adding AWGN to the shifted segment to form the 128-sample example. The AWGN power is assigned as a controlled percentage *P* of the template signal power, varying from 0 to 100%. Figure 3b illustrates random examples generated from each template signal.

Afterward, examples generated from each template are concatenated under various labels to form the synthetic dataset. Several synthetic datasets are created by changing the labeling technique and the number of examples generated from each template signal $N_{T}\left[i\right]$. Various labeling strategies are employed to label examples generated from each template. First, examples generated from each template are uniquely labeled as different classes, such that $N_{c}=N$. Second, several combinations of examples generated from different templates are grouped and labeled as a unique mixed class, such that $N_{c}<N$. In this set of experiments, $N_{c}$ is changed from 2 to *N*. The dataset size $N_{DS}$ is calculated as $N_{DS}=\Sigma_{i=0}^{N}N_{T}\left[i\right]$. The number of examples in each class (dataset balance) is controlled by the labeling strategy and the number of examples generated per template $N_{T}\left[i\right]$. The class distribution is varied from balanced, with an equal number of examples per class, to unbalanced, with a minority to majority class ratio of 10%. For example, a 4-class unbalanced dataset is created with the number of examples per class set to 10000, 5000, 2500, and 1250. The dataset is then randomly stratified and divided into training, validation, and testing sets distributed as 64%, 16%, and 20%, respectively. For some experiments, additional AWGN is added to the testing dataset only to investigate the model capability of handling noisy data. Table 2 depicts the synthetic dataset parameters used in this section.

#### 3.1.2. Experimental Setup and Tools

Thereafter, the proposed MF CNN classifier illustrated in Figure 2 is trained, validated, and tested using the synthetic dataset. Keras, with the Tensorflow backend, is used to build and train the MF CNN classifier on the synthetic dataset with various parameters. TensorFlow is an open-source framework for ML created by Google with a variety of tools and libraries that helps developers to build ML models. Keras is an open-source software library with a Python interface for the TensorFlow library [36].

Raw dataset examples are fed to the MF CNN model that produces the class probabilities as the model output. The Adam optimizer with adaptive rate scheduling is used for model training with an initial learning rate of 0.001 and a decay rate of 0.9. The loss function is set to the categorical cross-entropy loss, and the optimization objective is set to maximize the model classification accuracy of the multiclass problem at hand. The model is trained and validated using a minibatch size of 512, and the number of epochs is set to 200, with an early stopping callback tied to the validation loss. The training process is carried out on a cloud machine featuring 8 CPU cores, 30 GB of RAM, and an NVIDIA QUADRO RTX 5000 GPU with 16 GB of VRAM.

The number of Conv1D filters $N_{F}$ and the kernel size $N_{k}$ model parameters are set as controlled variables. Several model variants are built by modifying $N_{F}$, $N_{K}$, and controlling the model learnable layers and initialization methods. The Conv1D and FC output layers are set to either trainable or nontrainable, with different initialization methods and kernel weight constraints. For all model variants, the bias term of the Conv1D and FC output layers is set to zero to follow the MF operation. Table 2 depicts the model parameters used in these experiments. In the following, we focus, for brevity, on model variants with $N_{F}=N_{C}=N$, which results in an FC kernel with a square weight matrix of N×N size. However, interpretations provided in this section can be generalized to all model variants.

In the first model variant, both Conv1D and FC layers are set to nontrainable, the template signals are assigned to the Conv1D kernel weights such that each filter is assigned a unique template signal, while the kernel weights of the FC output layer are assigned a sparse identity square matrix with all weights set to zero, excluding the diagonal weights which are set to 1. The number of learned parameters of this model is minimized; only BN layer parameters are learned. For this model variant, the input signal is correlated with the MF templates embedded in the Conv1D layer filters, and the maximum activated output of each filter produced by the GPM layer is fed to the FC layer. The identity matrix assignment of the FC layer kernel is used to force the layer to apply a 1-to-1 input-to-output mapping in which each maximum correlation signal is mapped directly to the output probability of the corresponding class. The weighted sum of the FC layer inputs is not computed in this variant due to disabling layer training and the identity matrix assignment of the kernel weights. In this model variant, the MF operation is typically performed, and the classifier is expected to minimize loss, maximize accuracy, and minimize the training time.

In the second model variant, both Conv1D and FC layers are set to trainable, and the weights of both layers are initialized using the default Glorot uniform initializer. However, kernel weights of the FC layer are constrained to a sparse diagonal matrix with all elements set to zero, excluding the *N* diagonal weights which can be learned during model training. This constraint forces the FC kernel to map each layer input to only one output but allows the layer to learn mapping scales. For this model variant, the Conv1D layer kernel weights are learned to minimize the loss under the constraint of FC layer diagonal weights, which maps and scales the maximum output of each Conv1D filter to the corresponding input of the softmax activation function without computing the weighted sum of the FC layer inputs due to the 1-to-1 mapping imposed by the diagonal weight constraint. In this model variant, the Conv1D layer kernels are expected to learn the MF templates, while the FC layer is expected to learn optimal values of the kernel diagonal weights to minimize loss.

In the third model variant, both Conv1D and FC layers are set to trainable, and the weights of both layers are initialized using the default Glorot uniform initializer without applying weight constraints to any layer. For this experiment, the Conv1D layer kernel weights are trained under no constraints, and the N×N weights of the FC layer are learned. In this model variant, the Conv1D layer kernels are expected to learn the MF signal templates, while the FC layer kernel is expected to learn the optimal weighted sum of the layer inputs to compute class probabilities that minimize loss and maximize model accuracy. This model variant represents the most common way of training ML models in the literature, in which the model architecture is established by the designer, leaving the process of finding and optimizing model weights to the optimizer.

#### 3.1.3. Results and Visualizations

More than 100 experiments were conducted using various dataset and model parameters, which are carefully selected to validate our MF interpretation of the proposed classifier and to study the effect of noise, dataset size, dataset balance, labeling technique, and model parameters on the learning experience and performance of the proposed MF CNN classifier. In each experiment, the model is trained, validated, and tested using a variant of the synthetic dataset with specific parameters; the model learning curves and accuracy scores including classification accuracy, recall, precision, and F1 score are collected; and the kernel weights of the Conv1D and FC output layers are extracted from the trained model, visualized, and compared with the template signals. Unfortunately, the testing results of all experiments cannot be detailed in this space, yet we attempt to provide a concise summary and conclusions on the results.

At the performance level, the proposed MF CNN classifier achieves a 100% accuracy score on the training and validation sets for all conducted experiments with no exclusions. In all experiments, the optimizer can reach 100% validation accuracy in a few epochs (fewer than 20), where the epoch time depends on the dataset size; for example, the epoch training time is less than 1 s for $N_{DS}=$ 40,000. The total number of model parameters is in the range of hundreds depending on $N_{F}$, $N_{K}$, and $N_{C}$. The model testing accuracy score ranges from 90% to 100%, where the upper bound is achieved for most synthetic datasets with a typical common set of parameters, such as a balanced dataset with moderate size, whereas the lower bound is achieved for the following extreme dataset parameters: the dataset size is limited to 1875 examples, the synthetic dataset is unbalanced with a minority to majority class ratio of 1:10, 100% AWGN is added to generate the dataset examples from the template signals, and an additional 100% AWGN is added to the testing dataset only. Degradation in the classifier performance in this experiment is attributed to the covariate dataset shift between the training and testing sets induced by the extra noise added to the testing set. Eventually, the achieved results of these experiments depict the classification capabilities of the proposed model and establish model robustness under different conditions including noise, small dataset size, dataset imbalance, and mixed classes.

At the MF validation level, for all experiments with trainable model layers and no weight constraints, the Conv1D filter kernels learn the template signals to a great extent, while the FC layer learns the optimal kernel weights to minimize loss. In the following, we advance and analyze the Conv1D and the FC output kernel visualizations for a synthetic dataset with common typical parameters such as a balanced dataset, moderate dataset size of 40,000 examples, identical distributions of the training and testing sets (no extra noise is added to the test set), and model parameters of $N_{C}=N_{F}=N=4$ and $N_{K}=64$. Figure 4a illustrates the learned kernel weights versus the corresponding template signals. Figure 4b depicts a heatmap image of the learned FC output layer weights.

As depicted by Figure 4a, the learned kernels mostly match the corresponding template signals with slight variations in the scale and waveform. On the other hand, all kernel weights of the FC output layer are learned during the training process, as shown by Figure 4b. Analyzing the diagonal weights, which represent the 1-to-1 mapping between a specific output of the GMP layer (maximum of the MF kernel outputs) and the corresponding class probability output, is of special importance because it precisely characterizes the learning process. For example, diagonal weights (1, 1) and (3, 3) have higher amplitudes, and their corresponding filter kernel waveforms 1 and 3 better match the corresponding template signals, as shown by Figure 4a. We call this behavior strong and weak kernel learning of the template. All diagonal weights are positive, except weight (2, 2), which indicates that the model learns an amplitude-revered version of the corresponding template signal. This behavior is noticed in all experiments with a trainable Conv1D layer for one or more diagonal weights and can be attributed to the Tanh activation function, which has an output range from −1 to 1. The nondiagonal weights of the FC output layer exhibit the contribution of GMP layer outputs in computing the inactivated class probabilities. Eventually, the class probability is computed as a softmax-activated weighted sum of the GMP layer outputs.

#### 3.1.4. Analysis and Discussion

Finally, some observations drawn from the conducted experiments, which are not presented herein, are reported in the following. Models with nontrainable Conv1D and FC output layers with preassigned weights achieved the best scores and training time but required prior knowledge of the class templates, which is not common in ML problems. Models trained on datasets generated with small or zero noise-to-signal percentages *P* tended to learn weak kernels that do not precisely match the template signals. This is attributed to the model not being forced to learn strong kernel templates to minimize loss due to the easiness of classifying time-shifted versions of the template signals. This phenomenon is reversed for models trained using datasets generated with higher *P*.

On the other hand, models trained on unbalanced datasets tend to produce weak kernels for the majority classes and strong kernel templates for the minority class. This is attributed to classifying the minority classes being more difficult than the minority classes which requires the model to learn better representation (templates) of the class examples. The Conv1D kernels of models trained on mixed-class datasets (each class contains a mix of examples generated by different templates) learn the template signals of individual subclasses, and the FC output layer maps the GMP layer outputs to the corresponding superclass. Models trained on small datasets learn weak kernel templates for all classes due to the insufficiency of the training examples to learn strong kernels.

At the model level, increasing the number of filters $N_{F}$ beyond the number of template signals *N*, and increasing the Conv1D kernel size $N_{K}$ beyond the template signal size, does not enhance the model learning process, but it results in learning repeated or redundant versions of the template signals and unnecessarily increasing the number of model parameters. However, in most ML problems, the number of template signals (features) and their actual size (duration) are unknown beforehand; consequently, it is recommended to gradually tune-up these parameters to efficiently optimize the model performance without unnecessarily increasing the model size. The Conv1D Tanh activation layer is essential in the proposed MF model because it tends to smooth the learned Conv1D kernels to match the template signal. Other activation functions such as ReLU, Sigmoid, and linear activation have been investigated but none of them can replace the Tanh activation function; in most cases, the classifier ability to learn the template signal and classification performance is significantly affected by using other activation functions.

The visualizations and analysis provided in this section provide an experimental proof of concept of the MF interpretation of the proposed CNN classifier and help develop a better understanding of the CNN learning process. Moreover, this set of experiments draws some guidelines for developing highly accurate, computationally efficient CNN classifiers. The developed experiments can be extended to develop a clear interpretation of deeper CNN models and conducted for other time series datasets with different characteristics.

## 4. Human Activity Recognition Using the Matched Filter CNN Classifier

In our previous work [18], we advanced the MF-based CNN for ECG classification at the edge. ECG signals are time series signals with specific morphologies that are correlated with different classes of cardiac arrhythmias. The proposed classifier was developed for a univariate single-lead ECG signal. The achieved results encourage us to extend this work to multivariate time series classification problems in other domains. In this work, we investigate applying the proposed MF CNN classifier to the sensor-based HAR. HAR uses sensor readings that form a multivariate time series to classify human activity. The HAR problem is approached as a multiclass time series classification problem. HAR at the edge is an active research area due to the abundance of sensory data and the need for accurate cost-effective solutions to analyze these data while addressing the concerns of cloud inference, including privacy, availability, and latency. The work presented in this section is an extension of the work presented in the last section, for which the proposed classifier is validated on a practical multivariate time series classification problem.

### 4.1. Datasets

In this work, the CNN MF classifier is applied to three widely used HAR datasets: UCI-HAR, MotionSense, and WISDM-AR databases. Data in the selected datasets were collected using various sensor types and devices and feature different characteristics such as the dataset size, balance, and the number of sensors to exhaustively validate the proposed CNN classifier and investigate its generalization capabilities.

The UCI-HAR dataset [21] is a public database collected from recordings of 30 volunteers performing 6 daily activities while carrying a Samsung Galaxy S II smartphone with embedded inertial sensors. For this dataset, 3-axial linear acceleration and 3-axial angular velocity were captured at a rate of 50 Hz using the embedded accelerometer and gyroscope sensors of the smartphone. The experiments were video-recorded and manually labeled. The UCI-HAR dataset was randomly partitioned into training and testing sets of 70% and 30%, respectively. A noise removal filter was applied to the sensor data, which was segmented using a fixed-width sliding window of 2.56 s and 50% overlap (128 readings/segment). The sensor acceleration signal comprising gravitational and body motion components was separated using a low-pass filter with a 0.3 Hz cutoff frequency into body acceleration and gravity. Therefore, the total number of sensor signals (channels) in this dataset is 9.

The MotionSense dataset comprises time series data collected from a set of 24 participants performing 6 activities in 15 trials of the same environment and conditions [37]. Accelerometer and gyroscope sensor data were collected using an iPhone 6s smartphone and sampled at a 50 Hz rate. The data were segmented using a fixed-width sliding window of 2.56 s (128 readings/segment). The MotionSense Dataset is prepartitioned by trials into training and testing sets of 75% and 25% distribution, respectively. The total number of sensor channels in this dataset is 6.

In the WISDM-AR dataset [20], accelerometer data were collected and labeled from 29 subjects wearing a smartwatch and performing 6 daily activities at a rate of 20 Hz. The sensor signals were segmented using a fixed-width sliding window of 2.6 s and 25% overlap (52 samples/window). The overlap interval is deliberately limited to 25% (0.6 s) to yield mostly different examples instead of repeating shifted versions of the dataset examples as in some approaches which use highly overlapped segments. The total number of sensor channels in this dataset is 3.

The training and testing set class distribution is illustrated in Figure 5. Both UCI-HAR and MotionSense datasets have a preassigned division of the training and testing sets, whereas the WISDM-AR dataset is randomly stratified and partitioned into training, validation, and testing sets of 64%, 16%, and 24%, respectively. The UCI-HAR and WISDM-AR are small datasets with around 10 K examples, while the MotionSense dataset contains around 35 K examples. The UCI-HAR and MotionSense are balanced datasets, whereas the WISDM-AR is an unbalanced dataset with a 2.54% minority-to-majority ratio. The numbers of time series channels used in the WISDM-AR, MotionSense, and UCI-HAR datasets are 3, 6, and 9, respectively. The selected datasets feature various characteristics and can be used to exhaustively test and validate the proposed MF CNN classifier.

### 4.2. Multivariate MF CNN Classifier

The model shown in Figure 6 is the multivariate version of the MF CNN classifier proposed in Section 3. The sensor input segments $N_{sen}$ are fed to a linear Conv1D layer with kernel size = 1 to extend the number of channels as needed by applying a weighted sum on the sensor signals that produces $N_{ch}$ channels. A BN layer is employed to normalize the input data as a preparation for adding AWGN to sensor data. The GaussianNoise layer is instantiated at the training time only for two purposes: first, to enable learning strong MF kernel templates as instructed by Section 3; second, to work as a regularizer to avoid model overfitting and reduce the variance between training and testing results. The remaining layers are similar to the proposed univariate MF CNN classifier.

The main variation between the univariate and multivariate model versions is using the DepthwiseConv1D layer for the MF operation, in which each input channel is convolved with a different kernel (called a depthwise kernel) independent of other channels. Depthwise convolution splits the input into individual channels, convolves each channel with an individual depthwise kernel with depth_multiplier output channels, and concatenates the convolved outputs along the channels axis. Unlike regular 1D convolution implemented by the multichannel Conv1D layer, depthwise convolution does not mix information across different input channels; in other words, it does not conduct cross-channel operations.

The main parameters of the DepthwiseConv1D layer are the kernel size $N_{K}$ and depth_multiplier, which controls the number of output channels generated per input channel in the depthwise convolution operation. The depth_multiplier argument determines how many filters are applied to one input channel, which is equivalent to the number of filters parameter $N_{F}$ of the Conv1D layer. Using the DepthwiseConv1D layer has two advantages over using the regular Conv1D layer. First, this approach enables training the Conv1D kernel filters separately to extract MF templates of a channel signal independent of other channels and apply the matched filtering operation proposed in this work. Eventually, the FC layer will map the maximum correlation outputs of each filter to the probability output of the corresponding class and find their optimal weighted sum to minimize loss. Second, the computation load of the DepthwiseConv1D layer is significantly reduced compared with the corresponding load of the equivalent Conv1D layer with the same parameters due to reducing the number of multiplications required to perform convolution [38]. Reducing the computation load of the CNN classifier is of special importance for edge inference. It should be indicated that we investigated replacing the DepthwiseConv1D layer with the Conv1D or SeparableConv1D layers, yet the achieved results using these layers fall behind the results achieved by the Depthwise Conv1D layer by a significant margin at the levels of model complexity, classification accuracy, and real-time performance.

Model parameters include the number of samples per segment $N_{S}$, number of sensor signals $N_{sen}$, number of model channels $N_{ch}$, depth_multiplier $N_{F}$, kernel size $N_{K}$, and number of output classes $N_{C}$. The model parameters are systematically tuned to maximize the model classification accuracy for the given dataset, and the optimal model parameters are depicted in Figure 6. The input linear layer is only instantiated for the WISDM-AR dataset to extend the number of channels to 9 instead of the 3 sensor signals; this layer results in around 6% improvement in accuracy for the WISDM-AR dataset. The kernel size $N_{K}$ is set to the input segment size $N_{S}$ for all datasets, which maximizes the convolutional layer receptive field and produces the best classification results. This can be linked to the MF kernel being able to extract nonredundant templates for the whole input segment interval. The number of kernel filters per channel $N_{F}$ is gradually increased for each dataset to find its optimal value that maximizes the model accuracy. In the proposed MF CNN model, $N_{F}$ represents the number of template signals, commonly called features, which are extracted for each input channel independently of other channels due to using the DepthwiseConv1D layer. The $N_{F}$ values depicted in Figure 6 are the optimal values that maximize the accuracy for which any increase or decrease in these values would result in a degradation of the model performance.

### 4.3. Methods and Tools

Keras with the Tensorflow backend is used to train and test the MF CNN classifier on the selected datasets. All model parameters are set to trainable without imposing any constraints on the layer kernel weights to ensure that the proposed model can achieve optimal results without requiring field expertise concerning the dataset or model architecture. The categorical cross-entropy loss function and Adam optimizer with an initial learning rate of 0.001 and adaptive learning rate scheduling are used for model training. The epochs and batch size parameters are set to 500 and 512, respectively, with a callback to save the model with the best validation accuracy score to avoid overfitting. The model is trained on the cloud machine described in Section 3.

Thereafter, models with the best scores are then optimized for edge computation using the TensorFlow and TensorFlow lite (TFLite) optimization tools [39,40]. The optimized Tensorflow models are converted to TFLite models for deployment on the edge device. TFLite is a package of tools that enables on-device inference of ML models. This package is composed of a runtime engine for ML model inference computation on edge devices and a set of tools for transforming and optimizing Tensorflow models post-training for usage on mobile and embedded devices. The model can be converted directly without quantization from the base model to a 32-bit floating-point (Float32) TFLite edge model. TFLite supports other quantization techniques such as Float16, dynamic range quantization, and full-integer quantization that reduces the model size and enables its usage on various platforms with distinct architectures. Quantization refers to techniques for performing computations and storing weights at bit widths lower than the floating-point precision. Quantization allows for a more compact model representation, smaller memory footprint, faster inference, and less-demanding computation requirements, yet it comes at the expense of accuracy loss. In this work, only Float32 TFLite models are developed, which are inherently supported by most modern smartphones and do not degrade the model performance.

Finally, the optimized TFLite classifier models are exported to the edge device for testing and benchmarking. For this purpose, a Samsung Galaxy Note 10 Lite with an octa-core ARM Cortex-A55 processor and 8 GB of RAM was used. The ARM Cortex processor architecture inherently supports 32-bit integer and floating-point operations. The TFLite models are benchmarked using the TFLite android benchmark tools, which measure and calculate statistics for the model average inference time and overall memory usage. The number of threads used for running the TFLite interpreter on the edge device is set to 4. The model performance metrics, including accuracy and F1 score, were measured for all TFLite models as well as the model benchmarking metrics, including the model size, memory usage, and average inference time.

## 5. Results and Discussion

The developed models are tested and benchmarked on the cloud machine described in Section 3 and the android phone edge device. On the cloud, classification performance metrics, including accuracy and F1 score, are measured for the training, validation, and testing sets. On the edge device, in addition to the classification metrics, model benchmarking metrics including the average inference time, model size, and memory footprint, are measured. The model accuracy is defined as the percentage of true predictions to the total number of dataset examples. In terms of true positives (TP), true negatives (TN), false positives (FP), and false negatives (FN), accuracy is defined as Acc=(TP+TN)/(TP+TN+FP+FN). The F1 score is defined for individual classes as F1=2TP/(2TP+FP+FN), and the average F1 score is computed for all classes in the dataset.

Table 3 shows the training and testing results of the MF CNN model classifier on the cloud machine and edge device for the selected datasets. The training time of the proposed model was proportional to the dataset size and ranges from 150 s for the WISDM-AR dataset to 20 min for the MotionSense dataset. Such a short training time of the proposed model is expected due to reducing the model complexity and number of learned parameters. The difference between training and testing accuracy did not exceed 2% for all datasets, as illustrated by Table 3 and the training curves depicted in Figure 7d , which indicates that the model generalizes well for all HAR datasets.

Figure 7 shows the confusion matrices of the proposed MF CNN classifier for the testing sets of the selected datasets. For all datasets, the main source of model confusion is the similarity or correlation between sensor readings or waveforms of different activities. Such a correlation would result from the similarity between some human activities or interindividual variations between the dataset subjects. For example, in the UCI-HAR dataset, the model confuses Sitting and Standing activities, which have highly correlated sensor readings. Such a behavior is not only challenging for our model but also for other HAR models presented in the literature, and it can be treated as an irreducible error.

The proposed MF CNN classifier achieves superior testing accuracy and F1 score for all selected datasets regardless of the dataset conditions, which supports the MF interpretation of CNNs and establishes the model generalization capability. High testing accuracy scores indicate that the model performs well regardless of the dataset size, the number of input channels, and the input segment size, whereas high F1 scores indicate that the model performs well regardless of the dataset imbalance problem manifested in the WISDM-AR dataset. The achieved performance does not come at the expense of overcomplicating the model; the number of model parameters ranges from 22,383 to 37,566, which is significantly small compared with the existing HAR classification models presented in the literature. The number of parameters varies among different datasets due to various values of $N_{S}$, $N_{F}$, and $N_{K}$, which are imposed by the dataset features such as $N_{S}$ and the optimal model parameters found during the model parameter tuning process such as $N_{F}$.

In our experiments, we developed and tested deeper MF CNN models by hierarchically stacking multiple Conv1D and MaxPooling layers and carefully adjusting the pooling parameters and hierarchical kernel sizes to perform the matched filtering operation presented in this work. Some deeper CNN models can achieve better classification results at the expense of increasing the model size and computational complexity, which affects their suitability for edge inference. Therefore, deeper MF CNN models are not presented herein, and their results are not reported in this article.

Finally, we developed an android application for HAR using the TFLite models developed in this work and tested the app on a Samsung Galaxy Note 10 Lite smartphone. Figure 8 shows a screenshot of the developed app. The developed app is granted access to the accelerometer and gyroscope sensors, and it performs the following functions: reads the needed sensor data, samples sensor readings using the dataset sampling frequency, creates a tensor buffer that aggregates $N_{S}$ sensor readings to form the model input segments, invokes the TFLite interpreter to perform the inference procedure, and displays the output class probabilities. The app has been personally tested by performing different activities and checking the displayed recognized activity.

### Comparison with Related Work

In the following, the proposed CNN MF model is compared with the state-of-the-art HAR classification methods. Table 4 lists related works presented in Section 2, methods used in those models, and their achieved classification and benchmarking results. Specifically, we compare the models in terms of classification accuracy, average F1 score, number of parameters, and average inference time on the edge device. Unfortunately, model complexity and run-time benchmarking results are not reported in many related works, however, they can be inferred from the model topology and used methods.

The first conclusion drawn from Table 4 is that our model outperforms by a significant margin all HAR models in terms of the model complexity and inference time at the edge without affecting the model performance. This is attributed to the shallow depth of the MF CNN HAR model; it uses a single Conv1D layer with a large receptive field followed by BN, GMP, and FC layers, whereas other models have deeper structures and use hybrid layers such as Conv1D, LSTM, and GRU. Another consequence of using deeper NN topologies is exposing the model to overfitting, which affects the model performance. Unfortunately, the model size and real-time performance results are not commonly reported for many models to compare them with our results and show the superiority of our model.

At the classification performance level, our model outperforms most models listed in Table 4 and achieves comparable results to the state-of-the-art results. Figure 9 illustrates the classification accuracy and F1 score of all models listed in Table 4 for the UCI-HAR, MotionSense, and WISDM datasets. These results show that the expressiveness power of the proposed model is not affected by reducing the model depth, due to increasing the receptive field of the convolutional layer and following the matched filtering technique.

However, there are some models, such as those presented in [22,24,27,32], that achieve better accuracy or F1 scores for some datasets. The variance in accuracy and F1 score between our model and these models does not exceed 1%, which can be compensated by fine-tuning the model parameters or using deeper versions of the proposed CNN classifier as discussed in Section 4. Moreover, such a small variance can result from the training/testing set distributions and overlapping periods between sensor segments during the creation of the dataset examples from the sensor readings. Creating highly overlapped segments yields repeated examples with minor variations that can be distributed to both the training and testing sets and, consequently, result in illusory superiority.

The computational complexity of the models with better classification results is significantly higher than our model, which is evidenced by either comparing the number of the reported model parameters or by inspecting the model topology and used layers. The number of parameters of the model presented in [32] is 50 times larger than our model, while the models presented in [24,27] are DNNs with hybrid Conv1D, GRU, BiLSTM, and transformer layers, which increases the model complexity and computational cost.

On the other hand, there are some models, such as the model presented in [24], which outperform our model for only a specific dataset (WISDM-AR) but not the other (UCI-HAR). This observation demonstrates that the generalization capability of the proposed MF CNN classifier is better than most related models presented in the literature. This conclusion is also supported by the proposed MF CNN classifier achieving convergent classification performance for all selected HAR datasets regardless of the dataset characteristics.

Limitations of the proposed classifier can be divided into algorithmic and computational limitations. At the algorithmic level, the proposed classifier is based on the MF theory and, consequently, it suffers from MF difficulties in distinguishing highly correlated examples belonging to different classes. Such a limitation can be overcome by implementing deeper versions of the CNN model to increase model expressiveness and feature extraction capabilities. At the level of computation complexity, the proposed model is limited in terms of the number of layers (model depth) and input segment size to minimize the model complexity and computational cost. The proposed model is well-suited for short-term HAR applications, which are commonly available in the HAR literature, rather than long-term HAR problems. Both algorithmic and computational limitations can be overcome using deeper versions of the model, which can be developed and optimized for cloud inference.

Eventually, the proposed model outperforms all related models in terms of model complexity (number of parameters) and computational cost (inference time), while achieving comparable accuracy results. The achieved results of the proposed MF CNN classifier enable its deployment on a wide range of edge devices for real-time HAR.

## 6. Conclusions and Future Work

In conclusion, we presented a clear interpretation supported by an experimental proof of concept of the Conv1D CNN classifier operation as an MF. The visualizations and analyses provided in this work can help develop a better understanding of the CNN learning process. Moreover, the set of experiments carried out in this work draws some guidelines for developing highly accurate computationally efficient CNN classifiers.

Based on this interpretation, we proposed a novel HAR model optimized for edge deployment that can be embedded in a wearable device or mobile phone for activity recognition. The generalization capability of the proposed model was established by testing the model on three recognized HAR databases. The proposed model was thoroughly evaluated and benchmarked on an android device, and the results are reported and discussed. The proposed edge model is superlight in terms of real-time execution performance and model size, and it offers superior classification results compared with the state-of-the-art HAR models. A distinct feature of the proposed model is its readiness for deployment on tightly resource-constrained edge devices for real-time HAR. A very important conclusion drawn from this work is that interpretable development of neural networks leads to building more efficient and accurate models.

Limitations of the proposed MF CNN classifier include algorithmic limitations, due to the MF correlation operation and shallow depth of the CNN model, and computational constraints imposed by the edge inference approach. In our future work, we will attempt to overcome such limitations by using deeper versions of the MF CNN classifier and opting for the cloud inference approach if needed. Furthermore, we will attempt to extend the provided interpretation and investigate enhancing the classifier performance using deeper versions of the model. The MF CNN classifier model will be investigated for other relevant time series classification problems. Another promising research direction inspired by this work is investigating different signal-processing-based interpretations of the convolutional layer—convolution is a fundamental operation in signal processing and it constitutes the basis of many principles and applications in this domain—and using these interpretations to build better and more efficient models that push ML research boundaries.

## Figures and Tables

**Figure 1 sensors-22-08060-f001:**
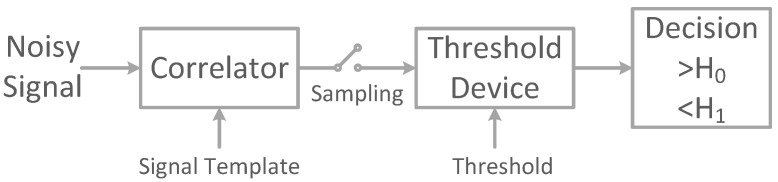
Block diagram of the matched filter receiver.

**Figure 2 sensors-22-08060-f002:**
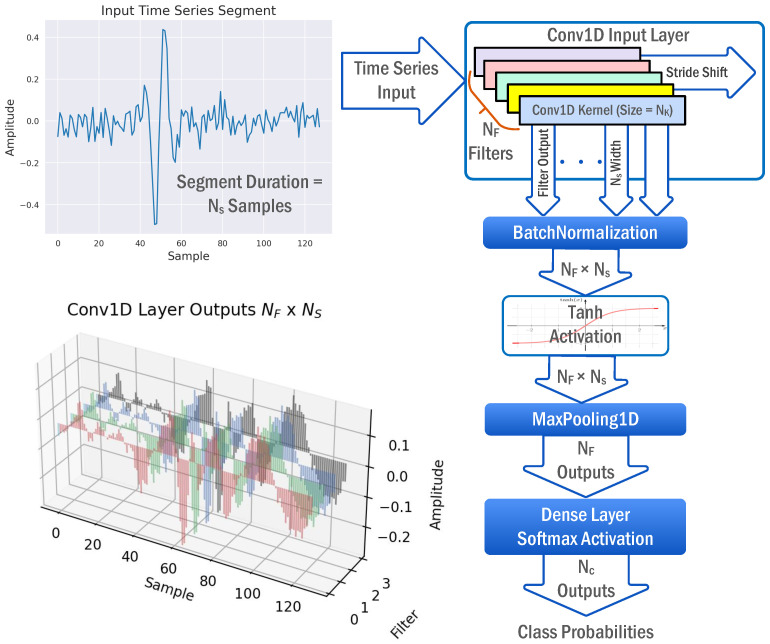
The MF CNN classifier model.

**Figure 3 sensors-22-08060-f003:**
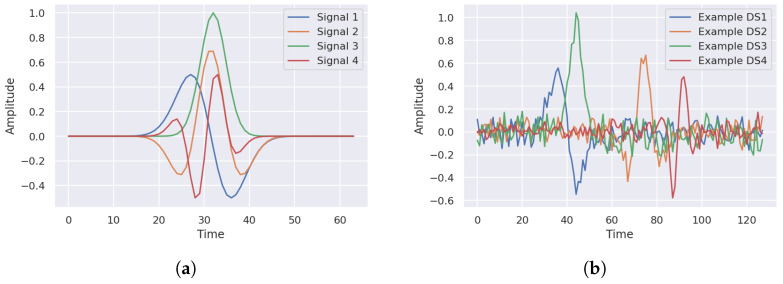
Template signals and random examples of the synthetic dataset. (**a**) Template signals used in generating the synthetic dataset (N=4). (**b**) Random signals of the synthetic dataset examples with %50 noise-to-signal percentage.

**Figure 4 sensors-22-08060-f004:**
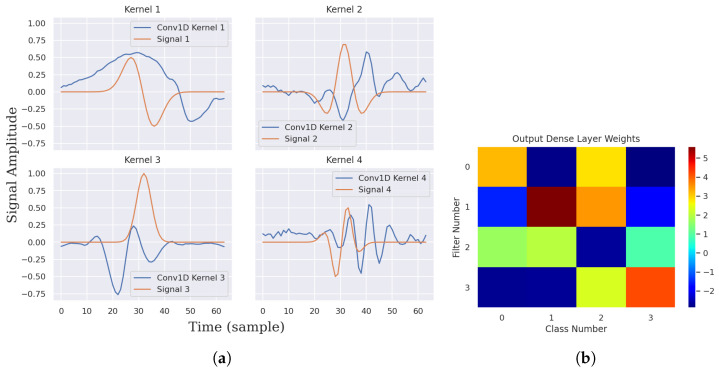
Learned Conv1D and FC output layer kernel weights for the synthetic dataset with N=4, $N_{T}\left[i\right]=10000\;\mathrm{for}\;i\in \left\{0,1,2,3\right\}$, P=50%, $N_{c}=4$, $N_{DS}=40,000$, balanced dataset, and the third model variant with $N_{F}=N_{c}=N=4$, $N_{k}=64$ , $N_{S}=128$. (**a**) Learned Conv1D filters versus template signals. (**b**) Learned kernel weights of the FC output layer.

**Figure 5 sensors-22-08060-f005:**
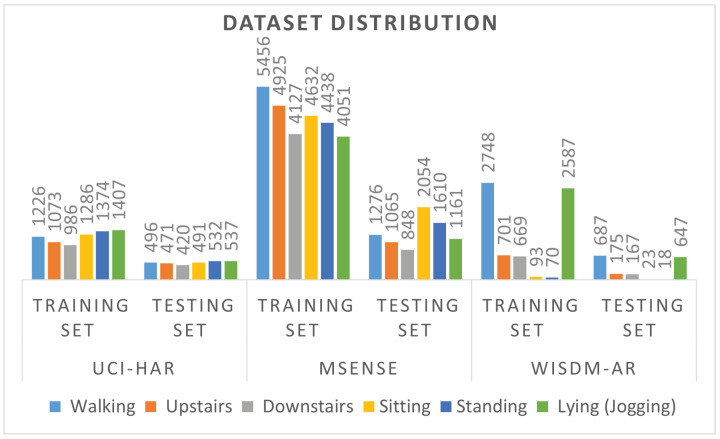
Training and testing set distribution of the selected datasets.

**Figure 6 sensors-22-08060-f006:**
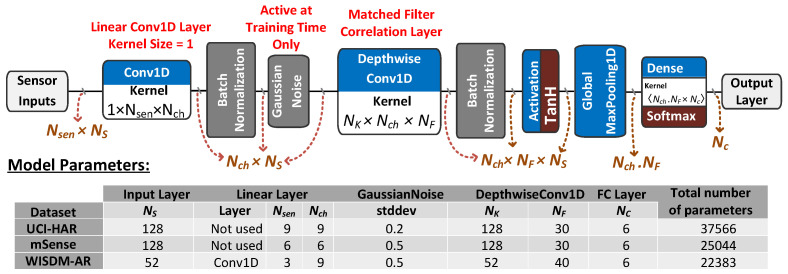
Multivariate MF CNN classifier.

**Figure 7 sensors-22-08060-f007:**
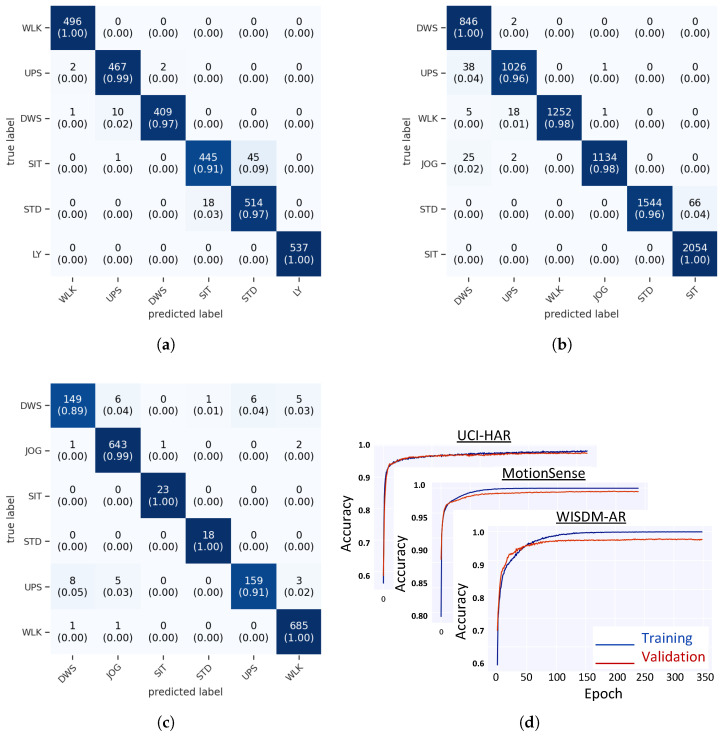
Confusion matrices of the CNN MF classifier for the test set of the selected datasets and the training curves of the proposed MF CNN classifier for the selected datasets. (**a**) UCI-HAR Confusion matrix. (**b**) MotionSense Confusion matrix. (**c**) WISDM Confusion matrix. (**d**) Training curves of the MF CNN classifier for all datasets.

**Figure 8 sensors-22-08060-f008:**
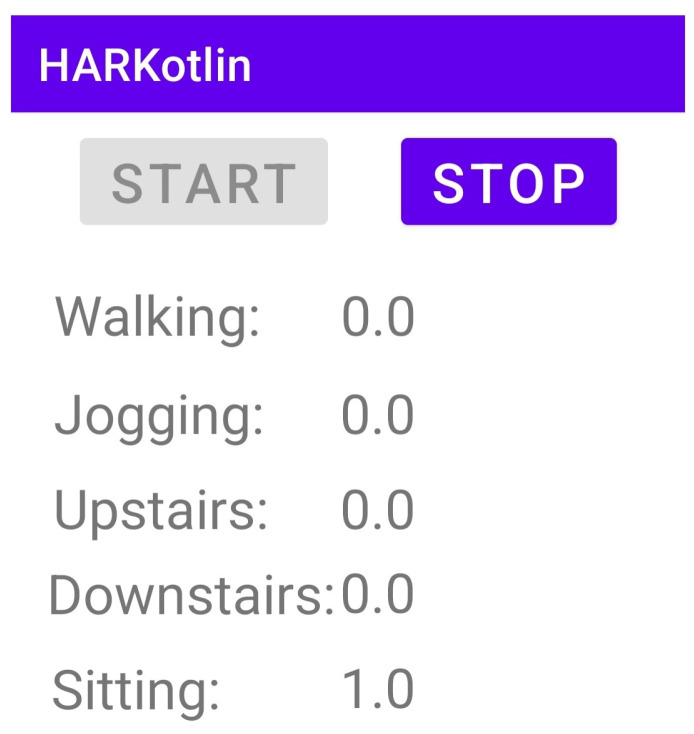
Screenshot of the developed HAR android application.

**Figure 9 sensors-22-08060-f009:**
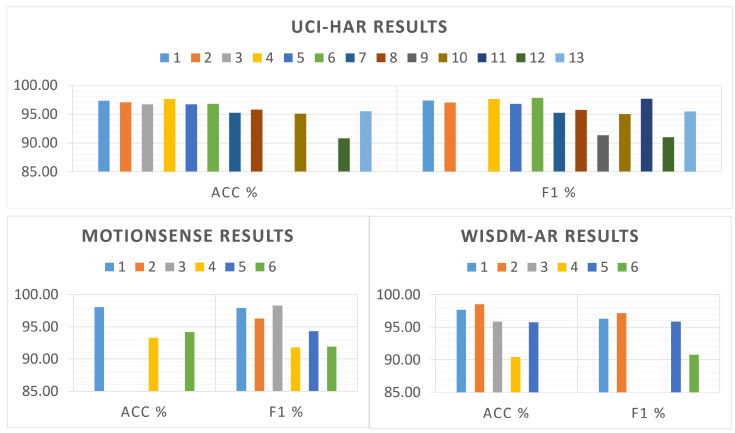
Comparing the proposed model accuracy and F1 score with related models.

**Table 1 sensors-22-08060-t001:** Summary of HAR-related work presented in this section.

Work	Used Methods	Limitations
Ignatov [22], 2018	CNN + Statistical Features	Statistical feature extraction requires additional computational cost
Xia et al. [23], 2020	CNN + LSTM	The model depth and layer diversity increases the model complexity
Nafea et al. [24], 2021	CNN + BiLSTM	
Yin et al. [25], 2022	CNN + BiLSTM + Attention	LSTM and GRU RNNs suffer from increased computation time, limiting their applicability to edge inference
Tan et al. [26], 2022	Conv1D + GRU + Ensemble learning	
Pushpalatha and Math [27], 2022	CNN + GRU+ FC	Models tested on a single dataset do not establish the model generalization capabilities
Sikder et al. [28], 2019	CNN	Using such a DNN increases the computational cost of the model
Luwe et al. [29], 2022	CNN + BiLSTM	Using a DNN model with hybrid layers increases the model complexity and computational cost of the proposed classifier
Ronald et al. [30], 2021	CNN + BiLSTM + Inception + ResNet	Such a deep model is not the best fit for edge inference, which requires smaller models with a reduced computational cost.
Sannara EK [32], 2022	CNN + Transformer	The number of parameters is greater than 1 million
Tang et al. [33], 2021	Teacher-Student CNN	
Rahimi Taghanaki et al. [34], 2021	CNN + FC + Transfer Learning	Results achieved by self-supervised and semisupervised models fall behind their supervised learning counterparts by a considerable margin
Taghanaki et al. [35], 2022	CNN + STFT + Transfer Learning	

**Table 2 sensors-22-08060-t002:** The dataset and model parameters used in this set of experiments.

	Symbol	Definition	Range of Values
**Dataset**	*N*	Number of template signals	2–5
	$N_{T}\left[i\right]$	Number of examples per *i*th temp	100, 1000, 10000
	*P*	Noise-to-signal %	0, 50%, 100%
	$N_{c}$	Number of classes	2–5
	$N_{DS}$	Dataset size	$\Sigma_{i=0}^{N}N_{T}\left[i\right]$
	Balanced	Is the dataset balanced	Yes, No
**Model**	$N_{F}$	Number of Conv1D layer filters	64, 128
	$N_{K}$	Conv1D kernel size	64, 128
	$N_{S}$	Model input size	128
	Learnable	Learn layer weights	True, False

**Table 3 sensors-22-08060-t003:** Training, validation, and testing results of the MF CNN classifier on the UCI-HAR, MotionSense, and WISDM-AR datasets.

Dataset	Training Results			Validation %		Testing %		Number of Params	Android Benchmarking Results (Float32)		
	Acc %	F1 %	Time (s)	Acc %	F1 %	Acc %	F1 %		Average Infer Time (μs)	Memory Footprint (MB)	File Size (KB)
**UCI-HAR**	98.98	97.32	523.63	97.82	97.69	97.32	97.35	37,566	672.60	2.98	149.75
**mSense**	99.88	99.88	1199.88	98.73	98.51	98.03	97.88	25,044	517.28	2.92	101.65
**WISDM-AR**	99.95	99.53	151.62	97.60	96.86	97.67	96.34	22,383	294.63	2.96	89.46

**Table 4 sensors-22-08060-t004:** Comparing the proposed model performance and benchmarking results with related works. Bold text indicates the best metric results per dataset.

	ID	Work	Used Methods	ACC %	F1 %	Number of Parameters	Inference Time (ms)
**UCI-HAR**	**1**	Proposed MF CNN	CNN	97.32	97.35	**37,566**	**0.67**
	**2**	Nafea et al. [24], 2021	CNN + BiLSTM	97.04	97.00	–	–
	**3**	Yin et al. [25], 2022	CNN + BiLSTM + Attention	96.71	–	–	14.71
	**4**	Ignatov [22], 2018	CNN + Statistical Features	**97.62**	97.63	–	–
	**5**	Tan et al. [26], 2022	Conv1D + GRU + Ensemble learning	96.70	96.80	–	1.68
	**6**	Pushpalatha and Math [27], 2022	CNN + GRU+ FC	96.79	**97.82**	–	–
	**7**	Sikder et al. [28], 2019	CNN	95.25	95.24	–	–
	**8**	Xia et al. [23], 2020	CNN + LSTM	95.80	95.78	49,606	–
	**9**	Tang et al. [33], 2021	Teacher-Student CNN	–	91.35	–	–
	**10**	Ronald et al. [30], 2021	CNN + BiLSTM + Inception + ResNet	95.09	95.00	1,327,754	–
	**11**	Sannara EK [32], 2022	CNN + Transformer	–	97.67	1,275,702	6.40
	**12**	Rahimi Taghanaki et al. [34], 2021	CNN + FC + Transfer Learning	90.80	91.00	–	–
	**13**	Luwe et al. [29], 2022	CNN + BiLSTM	95.48	95.45	–	–
**MotionSense**	**1**	Proposed MF CNN	CNN	**98.03**	**97.88**	**25,044**	**0.52**
	**2**	Tang et al. [33], 2021	Teacher-Student CNN	–	96.31	–	–
	**3**	Sannara EK [32], 2022	CNN + Transformer	–	98.32	1,275,702	6.40
	**4**	Rahimi Taghanaki et al. [34], 2021	CNN + FC + Transfer Learning	93.30	91.8	–	–
	**5**	Taghanaki et al. [35], 2022	CNN + STFT + Transfer Learning	–	94.30	–	–
	**6**	Luwe et al. [29], 2022	CNN + BiLSTM	94.17	91.89	–	–
**WISDM-AR**	**1**	Proposed MF CNN	CNN	97.67	96.34	**22,383**	**0.29**
	**2**	Nafea et al. [24], 2021	CNN + BiLSTM	**98.53**	**97.16**	–	–
	**3**	Yin et al. [25], 2022	CNN + BiLSTM + Attention	95.86	–	–	12.11
	**4**	Ignatov [22], 2018	CNN + Statistical Features	90.42	–	–	–
	**5**	Xia et al. [23], 2020	CNN + LSTM	95.75	95.85	49,606	–
	**6**	Tang et al. [33], 2021	Teacher-Student CNN	–	90.81	–	–

## Data Availability

The experiments have been carried out using public sensor-based HAR datasets including UCI-HAR [21], MotionSense [37], and WISDM-AR [20] which are open for use in the research work.

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
