# Peer review of "Matched Filter Interpretation of CNN Classifiers with Application to HAR"

_sensors, 2022, doi:10.3390/s22208060_

Round 1

Reviewer 1 Report

 Here, the authors presented a matched filter (MF) interpretation 3 of CNN classifiers associated with an experimental proof of concept using a carefully developed 4 synthetic dataset. they need to address the following comments

- many keywords given here. please make sure that 5 most used keywords enough

- The MF-based CNN model has been applied to the sensor-based human activity recognition (HAR) problem due to its significant importance in a broad range of applications. please mention at least one application here.

- there is no single reference used in the 3, 7, 8 paragraphs of the introduction section. check it and use if needed

- check line number 76 and 77, there is fullstop required after the statement.

-literature review done well about the CNN and human activity recognition work. But, please summarise at least few papers which are referred her in the table format by discussing the challenges and drawbacks of them. can be cited hand gesture classification using novel CNN crow search algorithm

- Conv1D input layer is looking clumsy and not easy to understand in the figure 2. please check it and modify

-briefly describe about the synthetic dataset with the parameters used

-which dataset used Keras with the Tensorflow backend is used to train and test the MF CNN classifier

-please compare the accuracy of the proposed model and other existing models and then show the comparison visually

Author Response

 Here, the authors presented a matched filter (MF) interpretation 3 of CNN classifiers associated with an experimental proof of concept using a carefully developed 4 synthetic dataset. they need to address the following comments

Response: We would like to thank the reviewer for his/her precious comments. We will do our best to address all his/her concerns.

- many keywords given here. please make sure that 5 most used keywords enough

Response:  We limited the keywords to: Machine Learning; Convolutional Neural Network; Interpretable Neural Network; Matched Filter; Human Activity Recognition

- The MF-based CNN model has been applied to the sensor-based human activity recognition (HAR) problem due to its significant importance in a broad range of applications. please mention at least one application here.

Response: In Section 2.2 Human Activity Recognition Related Work, the following sentence is already provided “HAR has become a very active research area due to its wide range of applications in elderly care, healthcare, smart homes, athletics, and abnormal activity monitoring.”

We added the following sentence and reference: “According to a 2021 World Health Organization (WHO) survey, over than 1 billion people live with some form of disability [19]. There are currently insufficient facilities to meet the requirements of people with disabilities. One of them is the requirement for a companion to monitor their activities. To safeguard people with disabilities from injury, danger, or accidents, they must be continually supervised and protected.”

  1. WHO. Disability and health. https://www.who.int/ne 1043 ws-room/fact-sheets/detail/disability-and-health, 2021. 1044 Accessed October 13, 2022.

- there is no single reference used in the 3, 7, 8 paragraphs of the introduction section. check it and use if needed

Response: We added reference [1] to Section 3 and 7.

Paragraph 8 introduces our work and thus no need to add references to it.

  1. Gu, J.; Wang, Z.; Kuen, J.; Ma, L.; Shahroudy, A.; Shuai, B.; Liu, T.; Wang, X.; Wang, G.; Cai, J.; et al. Recent advances in convolutional neural networks. Pattern recognition 2018, 77, 354–377.

- check line number 76 and 77, there is fullstop required after the statement.

Response: We corrected this typo and added the full stop.

-literature review done well about the CNN and human activity recognition work. But, please summarise at least few papers which are referred her in the table format by discussing the challenges and drawbacks of them. can be cited hand gesture classification using novel CNN crow search algorithm

Response: We added Table 1 to summarize the methods and limitations of the related HAR work. Unfortunately, the referred paper which addresses hand gesture recognition using images and 2D CNNs is not relevant to our work which addresses a completely different problem of HAR using sensor modalities. Therefore, there is no room for comparison or citing the paper as a related work.

- Conv1D input layer is looking clumsy and not easy to understand in the figure 2. please check it and modify

Response: We changed the Conv1D layer layout in Figure 2 to make it clear and easy to understand.

-briefly describe about the synthetic dataset with the parameters used

Response: Section 3.1.1 is completely devoted for describing the synthetic dataset and its parameters. Table 2 lists all parameters of the synthetic dataset. Figure 3 shows examples of the template signals used to generate the dataset and random examples of the generated signals.

-which dataset used Keras with the Tensorflow backend is used to train and test the MF CNN classifier

Response: Keras is used to train and test the proposed MF CNN classifier on both the Synthetic dataset and the three selected HAR datasets.

In Section 3.1.2 (Experimental Setup and Tools), the following is stated “Thereafter, the proposed MF CNN classifier illustrated in Figure 2 is trained, validated, and tested using the synthetic dataset. Keras with the Tensorflow backend is used to build and train the MF CNN classifier on the synthetic dataset with various parameters. TensorFlow is an open-source framework for ML created by Google with a variety of tools and libraries that helps developers to build ML models. Keras is an open-source software library with a Python interface for the TensorFlow library [36].”

In Section 4.3 (Methods and Tools), the following is stated “Keras with the Tensorflow backend is used to train and test the MF CNN classifier on the selected datasets. All model parameters are set to trainable without imposing any constraints on the layer kernel weights to ensure that the proposed model can achieve optimal results without requiring field expertise concerning the dataset or model architecture. The categorical cross entropy loss function and Adam optimizer with an initial learning rate of 0.001 and adaptive learning rate scheduling are used for model training. The epochs and batch size parameters are set to 500 and 512, respectively, with a callback to save the model with the best validation accuracy score to avoid overfitting. The model is trained on the cloud machine described in Section 3”.

-please compare the accuracy of the proposed model and other existing models and then show the comparison visually

Response: We compared the results of the proposed model and other related models in Section 5.1 (Comparison with Related Work). Table 4 shows the comparison results. We selected the tabular method which is commonly used in the literature to show the comparison results.

However, to address this concern we added a chart (Figure 9) to the same section to compare our model and related models in terms of accuracy and F1 score.

Reviewer 2 Report

The manuscript entitled “Matched Filter Interpretation of CNN Classifiers with Application to HAR” presents a matched filter (MF) approach to the convolutional neural network (CNN)-based time-series classifiers. The proposed approach allows for building light and highly-accurate classifier models. The proposed approach is validated on three datasets containing human activity recognition (HAR) data, achieving high accuracy and F1 score values. The method presented in the study outperforms the state-of-the-art HAR methods in terms of classification accuracy and run-time performance.

The manuscript is well-written and easy to follow. The theoretical background is scientifically sound and described in detail. The main contributions of the study are clearly stated. The methods are described in detail. The results are well presented (with room for some visual improvements) and appropriately discussed.

However, here are some comments I would like the authors to address before the manuscript is considered for publication:

1.      The literature review is well done, considering important and recent studies and placing the study within the narrow research field. This study applies CNN-based classification to 1D input time-series data. However, the application of CNNs with various 2D time-frequency signal representations has recently become a hot research topic, as the authors also mention one of their previous papers. Therefore, I would like to suggest the authors briefly discuss the advantages and disadvantages of this alternative approach compared to the one in the study in the Introduction section and also supplement the introductory part with some of the recent studies on this topic to briefly illustrate the state-of-the-art performances of the CNNs and time-frequency representations in time-series signal classification. Please consider briefly mentioning the following papers for illustration purposes: 10.1007/s10044-020-00921-5, 10.1109/ACCESS.2021.3139850, 10.1109/TNNLS.2020.3008938.

2.      Please move Figures 2, 6, and 8 below their respective paragraphs.

3.      Please increase the text size in Figure 7, as it is difficult to see without extensive zooming.

4.      In the Conclusions section, please address some of the limitations of the presented study.

5.      In the Conclusions section, please also elaborate more on the directions mentioned for future work.

Author Response

The manuscript entitled “Matched Filter Interpretation of CNN Classifiers with Application to HAR” presents a matched filter (MF) approach to the convolutional neural network (CNN)-based time-series classifiers. The proposed approach allows for building light and highly-accurate classifier models. The proposed approach is validated on three datasets containing human activity recognition (HAR) data, achieving high accuracy and F1 score values. The method presented in the study outperforms the state-of-the-art HAR methods in terms of classification accuracy and run-time performance.

The manuscript is well-written and easy to follow. The theoretical background is scientifically sound and described in detail. The main contributions of the study are clearly stated. The methods are described in detail. The results are well presented (with room for some visual improvements) and appropriately discussed.

Response: We would like to thank the reviewer for his/her precious comments. We will do our best to address all his/her concerns.

However, here are some comments I would like the authors to address before the manuscript is considered for publication:

  1. The literature review is well done, considering important and recent studies and placing the study within the narrow research field. This study applies CNN-based classification to 1D input time-series data. However, the application of CNNs with various 2D time-frequency signal representations has recently become a hot research topic, as the authors also mention one of their previous papers. Therefore, I would like to suggest the authors briefly discuss the advantages and disadvantages of this alternative approach compared to the one in the study in the Introduction section and also supplement the introductory part with some of the recent studies on this topic to briefly illustrate the state-of-the-art performances of the CNNs and time-frequency representations in time-series signal classification. Please consider briefly mentioning the following papers for illustration purposes: 10.1007/s10044-020-00921-5, 10.1109/ACCESS.2021.3139850, 10.1109/TNNLS.2020.3008938.

Response: We agree that the application of CNNs with various 2D time-frequency signal representations has recently become a hot research topic, and this method has been presented in our previous work. However, the context of this article neither allows comparing the time-frequency-based methods to the proposed classifier nor citing the referred papers.

The application targeted by this work is HAR and thus we limited related work to this application. Unfortunately, the papers referred by this comment address different applications (Laser Interferometer Gravitational-Wave Observatory (LIGO) detector, Speech Processing, and EEG). Therefore, there is no room for comparison at the application level.

We think that such a comparison is better made in a work that uses time-frequency representations for time series classification instead of our work that proposes a completely different interpretation of 1D CNNs. We plan to present other time-frequency-based classifiers in our future work, and we think that the suggested references will be helpful in that work.

  1. Please move Figures 2, 6, and 8 below their respective paragraphs.

Response: We moved the referred figures below their respective paragraphs as requested.

  1. Please increase the text size in Figure 7, as it is difficult to see without extensive zooming.

Response: We increased the size of Figure 7 to address this concern.

  1. In the Conclusions section, please address some of the limitations of the presented study.

Response: In the first version of the article, the following paragraph is already advanced about the limitations of the proposed classifier in Section 5.1 just before the conclusions section

“Limitations of the proposed classifier can be divided into algorithmic and computational limitations. At the algorithmic level, the proposed classifier is based on the MF theory and, consequently, it suffers from MF difficulties in distinguishing highly correlated examples belonging to different classes. Such a limitation can be overcome by implementing deeper versions of the CNN model to increase model expressiveness and feature extraction capabilities. At the level of computation complexity, the proposed model is limited in terms of the number of layers (model depth) and input segment size to minimize the model complexity and computational cost. The proposed model is well suited for short-term HAR applications, which are commonly available in the HAR literature, rather than long-term HAR problems. Both algorithmic and computational limitations can be overcome using deeper versions of the model which can be developed and optimized for cloud inference.”

We added the following paragraph to the conclusions and future work section

 “Limitations of the proposed MF CNN classifier include algorithmic limitations due to the MF correlation operation and shallow depth of the CNN model, and computational constraints imposed by the edge inference approach. In our future work, we will attempt to overcome such limitations by using deeper versions of the MF CNN classifier and opting for the cloud inference approach if needed.”

  1. In the Conclusions section, please also elaborate more on the directions mentioned for future work.

Response: We added the following paragraph to the conclusions and future work section to address this concern.

"Furthermore, we will attempt to extend the provided interpretation and investigate enhancing the classifier performance using deeper versions of the model. The MF CNN classifier model will be investigated for other relevant time-series classification problems. Another promising research direction inspired by this work is investigating different signal processing-based interpretations of the convolutional layer —convolution is a fundamental operation in signal processing and it constitutes the basis of many principles and applications in this domain— and using these interpretations to build better and more efficient models that push ML research boundaries.”

Round 2

Reviewer 2 Report

The authors have addressed my comments.